# CRUST-Bench: A Comprehensive Benchmark for C-to-safe-Rust Transpilation

**Anirudh Khatry**♠     **Robert Zhang**♠*     **Jia Pan**♠*     **Ziteng Wang**♠*
**Qiaochu Chen**◇     **Greg Durrett**♠     **Isil Dillig**♠

♠ The University of Texas at Austin      ◇ New York University
akhatry@cs.utexas.edu

## Abstract

C-to-Rust transpilation is essential for modernizing legacy C code while enhancing safety and interoperability with modern Rust ecosystems. However, no dataset currently exists for evaluating whether a system can transpile C into *safe* Rust that passes a set of test cases. We introduce CRUST-Bench, a dataset of 100 C repositories, each paired with manually-written interfaces in safe Rust as well as test cases that can be used to validate correctness of the transpilation. By considering entire repositories rather than isolated functions, CRUST-Bench captures the challenges of translating complex projects with dependencies across multiple files. The provided Rust interfaces provide explicit specifications that ensure adherence to idiomatic, memory-safe Rust patterns, while the accompanying test cases enforce functional correctness. We evaluate state-of-the-art large language models (LLMs) on this task and find that safe and idiomatic Rust generation is still a challenging problem for various state-of-the-art methods and techniques. We also provide insights into the errors LLMs usually make in transpiling code from C to safe Rust. The best performing model, OpenAI o3, is able to solve only 19 tasks in a single-shot setting. Improvements on CRUST-Bench would lead to improved transpilation systems that can reason about complex scenarios and help in migrating legacy codebases from C into languages like Rust that ensure memory safety. [1]

## 1 Introduction

Code translation is essential for modernizing legacy systems, enabling cross-platform development, and improving software security (U.S. Government Accountability Office, 2022; Alexandrova et al., 2015; Egan, 2022; Giudice, 2024). As a memory-safe alternative to C, Rust has gained widespread adoption due to its strong compile-time guarantees that eliminate entire classes of memory bugs without relying on garbage collection. Major companies such as Google, Microsoft, and Amazon have integrated Rust into their infrastructure, and open-source efforts like the Linux kernel and WebRender have embraced it to reduce memory safety vulnerabilities.

However, translating C to Rust is not just about achieving functional equivalence: it also involves transitioning from non-memory-safe C semantics to memory-safe, idiomatic Rust. While Rust supports unsafe code for low-level operations, the core value of migration lies in producing code that compiles and executes within Rust's safe subset, allowing users to benefit from Rust's guarantees. Given that much of today's critical infrastructure is still written in C, there is a pressing need for reliable *automated* techniques that support not only C-to-Rust translation, but more importantly, C-to-*safe*-Rust migration.

Despite rapid progress in large language models (LLMs) for code generation, fully automated C-to-safe-Rust transpilation remains an open challenge. Existing ML benchmarks

---

[*]Authors contributing to dataset annotation.
[1]Code and Data available at https://github.com/anirudhkhatry/CRUST-bench

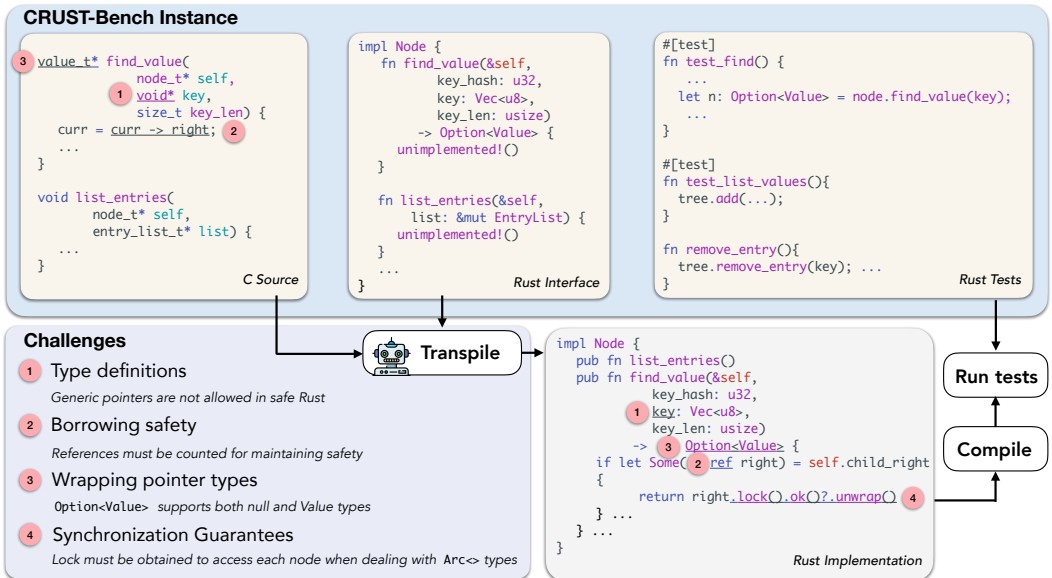

Figure 1: Example of a CRUST-Bench task: `btree-map`. **Top:** The task specification provided by CRUST-Bench, including the C source code (left), a safe Rust interface (middle), and Rust test cases (right). The C code implements the `find_value` function, which traverses a B-tree map to locate the value for a given key. This implementation relies heavily on raw pointers (e.g., `key`). In contrast, the Rust interface uses safe, structured types such as `Vec<u8>`, requiring the transpiler to generate memory-safe, idiomatic Rust. **Bottom right:** The expected Rust implementation, representing the actual target of the transpilation task. **Bottom left:** Additional challenges of the transpilation task are highlighted, illustrating the complexity of translating low-level pointer operations to safe abstractions.

largely focus on competitive programming problems (Chen et al., 2021; Austin et al., 2021; Hendrycks et al., 2021; Jain et al., 2025; Quan et al., 2025), which emphasize isolated, self-contained tasks rather than realistic, multi-file systems programming. Other efforts like SWE-bench (Jimenez et al., 2024) target bug-fixing scenarios with localized edits, rather than whole-program translation or structural refactoring. Prior benchmarks for C-to-Rust transpilation have similarly been limited, typically focusing on individual functions without testing for correctness at the integration level, and with little or no assessment of whether the resulting Rust code is safe or idiomatic. We elaborate on these limitations in Section 2.

To address these limitations, we introduce CRUST-Bench, a benchmark for evaluating automated C-to-Rust transpilation in realistic settings. CRUST-Bench is comprised of 100 C repositories, each paired with manually crafted Rust interfaces and test cases to assess the correctness and quality of transpilation. The Rust interfaces serve as formal specifications, defining function signatures, type annotations, and ownership constraints to guide the translation process. These interfaces enforce idiomatic and memory-safe Rust patterns, ensuring that transpiled code adheres to Rust's safety guarantees. The accompanying test cases provide an additional layer of validation, verifying that the generated Rust code correctly implements the specified behavior. Figure 1 provides a concrete example of a CRUST-Bench task, illustrating the gap between low-level pointer-based C code and the corresponding safe Rust implementation.

CRUST-Bench is built through a hybrid annotation process that combines automated tooling with human expertise. Annotators manually author safe and idiomatic Rust interfaces, along with corresponding test cases, for each C project. These interfaces are then validated via type checking using the Rust compiler to ensure they are well-formed and statically sound. This process ensures that CRUST-Bench provides a rigorous framework for assessment.

Using CRUST-Bench, we evaluate 12 frontier LLMs and other models on the task of C-to-Rust transpilation. The strongest frontier reasoning models such as OpenAI o3, Claude

| Benchmark | # Projects | Multi-file? | Avg LoC | Rust Interface? | Rust Tests? |
|---|---|---|---|---|---|
| CROWN | 20 | ✅ | 31.7K | ❌ | ❌ |
| TransCoder-Rust | 520 | ❌ | 108 | ❌ | ❌ |
| FLOURINE | 112 | ❌ | 68 | ❌ | ❌ |
| C2SaferRust | 7 | ✅ | 9.3K | ❌ | ❌ |
| SYZYGY | 1 | ✅ | 2.5K | ❌ | ❌ |
| LAERTES | 17 | ✅ | 24K | ❌ | ❌ |
| **CRUST-Bench (Ours)** | **100** | ✅ | **958** | ✅ | ✅ |

Table 1: Comparison of C-to-Rust transpilation benchmarks.

Opus 4, OpenAI o1 and Claude 3.7 Sonnet perform the best, successfully transpiling 13-22% of tasks in the one-shot setting and 32-48% of tasks when using a repair loop. The strongest open-source system is Virtuoso-Medium-32B (Arcee.ai, 2025), which outperforms the previous state-of-the-art distilled Arcee Nova-32B model, although it underperforms the closed-source models substantially. We also employ agentic systems such as SWE-agent (Yang et al., 2024c) and find that they do not outperform a "generate-then-repair" loop. Our error analysis highlights key challenges and directions for future work in LLM-powered transpilation.

Our primary contributions are: (1) We introduce CRUST-Bench, a new benchmark that enables systematic evaluation of C-to-Rust transpilation by incorporating repository-scale projects, annotated Rust interfaces, and correctness-enforcing test cases. (2) We provide an empirical analysis of the performance and limitations of frontier LLMs on this task, identifying key challenges for future research in automated code migration.

## 2 Motivation and Related Benchmarks

A number of benchmarks have been proposed to evaluate source code translation and transformation tasks, including transpilation (Sun et al., 2024), automated debugging (Li et al., 2024b; Jimenez et al., 2024), and safe code generation (Nitin et al., 2025; Eniser et al., 2024). While these benchmarks provide valuable insights, those that focus specifically on C-to-Rust transpilation tend to exhibit important gaps—in particular, limited task scope, lack of robust correctness validation, and insufficient evaluation of memory safety. Many are restricted to small, synthetic examples or overlook whether the generated Rust code adheres to safe and idiomatic practices. Table 1 summarizes the defining features of the most relevant existing benchmarks.

CROWN (Zhang et al., 2023) includes 20 C programs that are syntactically transpiled with guidance from Rust's ownership model. While the dataset features multi-file programs, it lacks structured interfaces or annotations to facilitate the generation of safe and idiomatic Rust code. TransCoder (Sun et al., 2024) is a widely used benchmark for training and evaluating neural machine translation models for code. For the C-to-Rust translation task, recent work (Yang et al., 2024a) typically uses a subset of 520 C functions. However, these are primarily small, self-contained units and do not represent full projects with complex dependencies or rich interfaces. FLOURINE (Eniser et al., 2024) evaluates large language models (LLMs) across 112 tasks derived from two larger real-world C projects. Nonetheless, the individual tasks remain narrow in scope and lack the structural complexity of complete software systems. C2SaferRust (Nitin et al., 2025) introduces a curated dataset of seven large-scale C programs (averaging 9.3K lines of code), each translated to Rust using a combination of LLMs and syntax-driven transpilation tools. Despite the scale of the individual programs, the overall dataset size is too limited to serve as a comprehensive benchmark for general-purpose evaluation. SYZYGY (Shetty et al., 2024) offers a single, large benchmark project of approximately 2,500 lines of code, fully transpiled to Rust, emphasizing correctness and idiomatic usage. LAERTES (Emre et al., 2021) provides a small collection of C programs for evaluation but lacks corresponding Rust implementations and interface specifications, limiting its utility for assessing safe or idiomatic transpilation.

Compared to prior benchmarks, CRUST-Bench includes a substantial number of real-world projects while keeping them within reach for LLM-guided transpilation techniques. With 100 repositories and an average size of 958 lines of code, it captures the challenges of migrating real-world software without requiring models to handle excessively large codebases. A key feature of CRUST-Bench is that it provides safe Rust interfaces and corresponding test files. This is important because existing transpilation tools, such as c2rust (Immutant Inc.), often produce naive translations that rely heavily on unsafe constructs, sidestepping Rust's safety guarantees rather than properly adapting C code. LLM-based approaches also risk generating inconsistent interfaces across files, making it difficult to produce a coherent, working Rust codebase. By defining explicit Rust interfaces and tests, CRUST-Bench forces models to generate code that integrates cleanly with a well-defined target.

## 3 Benchmark Structure

Each instance in our benchmark is built on a C repository, which we formalize as a collection of source files $\mathbf{S} = \{S_1, \ldots, S_n\}$, where $S_i$ represents an individual C source file. The goal of transpilation is to produce a corresponding Rust repository $\mathbf{R} = \{R_1, \ldots, R_n\}$ with a parallel file structure, ensuring that each $R_i$ is a direct translation of the corresponding $S_i$.

Validation of the transpiled Rust code is based on three criteria. First, it must conform to a well-defined interface $I = ((s_1, \ldots, s_n), (f_1, \ldots, f_m))$, which specifies a set of $n$ abstract datatypes (e.g. struct, enums) and $m$ functions. These interfaces enforce type constraints and function signatures to steer C-to-Rust transpilers to produce code that follows Rust's safety and ownership model. Second, the transpiled Rust code must compile without errors, meaning it must successfully type check and pass all borrow checker requirements enforced by the Rust compiler. Third, we provide a set of tests $\mathbf{T} = (t_1, \ldots, t_k)$, where each test is a function $t_i(\mathbf{R}) \rightarrow \{\mathsf{pass}, \mathsf{fail}\}$ that checks whether the transpiled code satisfies functional correctness requirements. Transpilation is considered successful if (1) $\mathbf{R}$ conforms to $I$, meaning all required functions are implemented with the appropriate types and ownership semantics, (2) $\mathbf{R}$ builds successfully and (3) all tests pass, i.e., $t_i(\mathbf{R}) = \mathsf{pass}$ for all $i$.

**Benchmark sourcing.** The projects in our benchmark are sourced from open-source repositories on GitHub, covering a diverse range of software categories. We consider repositories created between 2005 and 2025 that successfully compile using GCC 11.4.0 and Clang 14.0.0. As shown in Figure 2, CRUST-Bench includes projects from several domains, such as programming language infrastructure (e.g., compilers), algorithmic libraries (e.g., solvers), system utilities (e.g., shell implementations), and others.

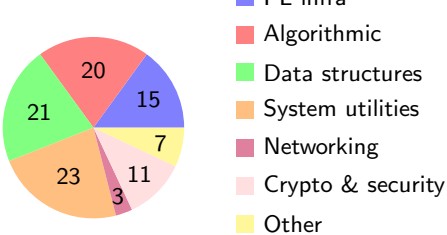

Figure 2: Application types.

**Preprocessing.** To construct CRUST-Bench, we apply a multi-stage preprocessing and selection pipeline. First, we filter for projects that are entirely written in C and meet a minimum complexity threshold: they must contain at least one dynamic memory allocation keyword, include build scripts, and have associated test files. We also perform automated deduplication to eliminate near-identical repositories. Following this initial filtering, we manually review

| C Code Properties | Avg | Max |
|---|---|---|
| Test cases | 76.4 | 952 |
| Test files | 3.0 | 19 |
| Test coverage | 67% | 100% |
| Lines of code | 958 | 25,436 |
| Pointer dereferences | 264 | 12,664 |
| Functions | 34.6 | 418 |

Table 2: Properties of C code

the remaining projects to assess overall code quality and ensure test completeness—for example, verifying that critical source or header files are not missing. We also exclude architecture-specific repositories, selecting only those that can run on both x86 and x64 systems to maintain portability. Finally, we omit GUI-based projects, as they often rely on external dependencies, platform-specific event loops, or graphical toolkits that are not easily testable in a headless, compiler- and test-driven evaluation setting.

Table 2 provides statistics about the C projects selected for CRUST-Bench. These projects exhibit a wide range of codebase sizes, with an average of approximately 958 lines of code. 60% of the projects in CRUST-bench contain a single source file with corresponding header files and test cases, and the remaining 40% projects contain more than one source file with associated header files and test cases. The average test coverage[2] is 67%, showing a moderate level of testing across the projects.

**Annotation process.** A team of four annotators who are authors of this paper manually defined the Rust interface *I* for each benchmark, ensuring that function signatures and data structures are both safe and idiomatic in Rust. The first annotation step is to convert C custom types, such as structs and enums, into their Rust equivalents. Annotators ensure that all types are native to Rust and adhere to its safety guarantees, avoiding unnecessary `unsafe` constructs and enforcing Rust's ownership model. After defining the types, they specify function signatures, including return types and ownership annotations. Instead of providing implementations, they replace function bodies with `unimplemented!()` to allow the Rust compiler to validate the interface independently.

Using these interfaces, annotators then construct Rust test files by adapting existing C tests. The test code invokes the corresponding Rust functions, ensuring that it correctly uses the defined interface. While the test files may include `unsafe` code for validation purposes, the core interfaces remain safe and idiomatic.

Finally, annotators compile the annotated Rust code to verify that all function signatures and test invocations are well-formed. On average, each annotator spent around an hour and a half per benchmark, and the first author manually reviewed all benchmarks for correctness. To check if the interfaces were implemented correctly, we did a pilot study where we implemented 20 benchmarks adhering to the interfaces and ran tests to see if they passed. We sampled long and short single-file projects, as well as projects containing multiple files with inter-dependencies. We also ensured that we included one sample from each domain from Figure 2. The annotators were able to successfully implement the projects with the desired functionality and pass all provided test cases.

**Evaluating the dataset.** Our final benchmark consists of 100 C projects annotated with corresponding Rust interfaces, capturing a wide spectrum of interface complexity and idiomatic Rust features, as summarized in Table 3. The dataset includes a total of 3,085 interface functions across 299 interface files, with an average of 30.9 functions per project. Many of these functions involve rich type-level structure: 44% use custom types as arguments, and 50% return custom types, reflecting a reliance on user-defined data structures. These types often require associated methods and trait implementations to enable idiomatic usage within Rust. Ownership is also prominently featured—56% of functions use reference arguments, and 30% involve mutable references—demanding precise handling of Rust's borrowing and mutability semantics.

| Metric | Total | Avg | Max |
|---|---|---|---|
| *Interface Structure* | | | |
| Interface files | 299 | 3.0 | 21 |
| Interface functions | 3,085 | 30.9 | 415 |
| Function arguments | 5716 | 57.2 | 1484 |
| *Ownership and Type Features* | | | **Percent** |
| % Functions with reference args | | | 56% |
| % Custom types in arguments | | | 44% |
| % Custom types in return types | | | 50% |
| % Functions with mutable references | | | 30% |

Table 3: Statistics of the Rust interface.

## 4 Experimental Setup: Models and Systems Evaluated

**Prompted LLMs.** We evaluate both closed-source and open-weight large language models (LLMs) on the CRUST-Bench benchmark. For closed-source models, we include o3, o1, o1-mini, and GPT-4o from OpenAI; Claude Opus 4, Claude-3.7-Sonnet and Claude-3.5-Sonnet from Anthropic; and Gemini-1.5-Pro from Google. On the open-weight side, we evaluate models with strong code generation capabilities: Virtuoso (Arcee.ai, 2025), a distilled variant

---

[2]We compute code coverage metrics over 25 repositories and discuss challenges of computing code coverage in Appendix A.

of DeepSeek-V3; QwQ-32B -Preview(Yang et al., 2024b); Qwen-Coder-32B-Instruct(Yang et al., 2024b); Deepseek-Coder-33B-Instruct(Guo et al., 2024); and LLaMA-3-8B (Llama-3 Team, 2024). Open-weight models are executed using the vLLM inference framework (Kwon et al., 2023) on a server equipped with four NVIDIA A40 GPUs.

For models that support temperature control (e.g., GPT-4o, Claude, Gemini), we use greedy decoding ($T = 0$) to ensure deterministic outputs and consistent pass@1 evaluation. Empirically, we found that higher temperatures significantly degraded performance, likely due to the long-context requirements and stringent correctness constraints of the tasks. On average, each model generates 5,165 tokens per task.

We experimented with variations in prompt formatting to identify the most effective strategy. The best results were obtained using a general task description followed by concise, point-based instructions. Notably, explicit directives—such as "do not use `libc`" and "the code must compile"—proved more effective than vague statements like "generate safe Rust." Each model is provided with the full C source file, the corresponding header file (if available), and the Rust interface specification and tests. We provide the prompts in Appendix C.

**LLM Scaling via Self-Repair.**   We investigate two strategies for leveraging additional compute to improve model performance through iterative self-repair. Both approaches begin with an initial model-generated candidate and refine it over multiple rounds based on feedback. The first strategy, referred to as **Compiler repair**, incorporates only compiler error messages into the prompt during each repair round. This allows the model to iteratively address syntactic and type-level issues flagged by the Rust compiler. The second strategy, called **Test repair**, extends this by also including information about failing test cases in the prompt, providing richer feedback about the correctness of the generated code.

For both strategies, we perform three rounds of self-repair per task. This bounded budget balances the benefits of iterative refinement with practical runtime constraints. In a pilot study on 10 benchmarks, we evaluated different configurations and found that starting with a single initial generation and applying greedy decoding across all repair rounds consistently outperformed sampling-based strategies, including the approach proposed by Olausson et al. (2024). In particular, we observed that deterministic repair with feedback led to more stable improvements, especially under the long-context, high-precision demands of the C-to-Rust transpilation task. Based on this finding, we adopt the single-candidate greedy repair strategy for our full evaluation. We describe further in Appendix B.

**Pipelined SWE-agent.**   Recent work on agent-based systems, such as SWE-agent (Yang et al., 2024c), has demonstrated strong capabilities in iteratively debugging and refining code by leveraging compiler feedback within isolated development environments. Notably, the combination of SWE-agent and Claude-3.7-Sonnet achieves state-of-the-art performance on the Full SWE-bench dataset as of May 2025.[3] Although these systems are primarily designed to address GitHub issues or apply localized patches in large codebases, their ability to interpret and act on compiler and testing errors makes them a promising candidate for supporting C-to-Rust transpilation.

To evaluate this potential, we adapt SWE-agent into a two-stage workflow we refer to as *pipelined SWE-agent*. In this setup, an LLM (e.g., Claude or GPT-4o) first generates an initial Rust implementation from a C source file and its associated Rust interface. SWE-agent then attempts to repair any resulting compiler errors by iteratively editing the generated code. To support this workflow, we configure SWE-agent with a Docker environment that includes the Rust toolchain (`cargo`, `rustc`) and Python, and we supply a custom problem specification and demonstration tailored to transpilation, following guidance from the SWE-agent documentation.[4] All other parameters are left at their default values. This design allows us to evaluate SWE-agent as a targeted post-processing and repair mechanism applied on top of LLM-generated code.

We also investigated the feasibility of using SWE-agent (Yang et al., 2024c) and OpenHands' CodedAct agent (Wang et al., 2025) for full end-to-end transpilation—i.e., supplying a C

---

[3]https://www.swebench.com/#test
[4]https://swe-agent.com/latest/config/demonstrations/

| Model | Pass@1 | | Pass@1 + Compiler repair (r=3) | | Pass@1 + Test repair (r=3) | |
|---|---|---|---|---|---|---|
| | Build | Test | Build | Test | Build | Test |
| OpenAI o3 | 35 | 19 | 68 | 31 | 63 | 48 |
| Claude Opus 4 | 43 | 22 | 78 | 29 | 65 | 40 |
| OpenAI o1 | 32 | 15 | 69 | 28 | 54 | 37 |
| Claude 3.7 | 26 | 13 | 54 | 23 | 49 | 32 |
| Claude 3.5 | 26 | 11 | 49 | 21 | 38 | 24 |
| o1-mini | 19 | 9 | 47 | 16 | 27 | 21 |
| GPT-4o | 18 | 7 | 52 | 18 | 42 | 22 |
| Gemini 1.5 Pro | 11 | 3 | 35 | 11 | 30 | 14 |
| Virtuoso (Distilled Deepseek V3) | 2 | 2 | 21 | 6 | 10 | 6 |
| Deepseek-Coder-32B | 1 | 0 | 2 | 0 | 2 | 0 |
| QwQ-32B-Preview | 1 | 0 | 1 | 0 | 1 | 0 |
| Qwen-2.5-Coder-32B | 0 | 0 | 0 | 0 | 0 | 0 |
| Adapted SWE-agent (Claude-3.7) | 41 | 32 | – | – | – | – |

Table 4: Pass rates on CRUST-Bench for different models in single-shot and repair settings.

repository as input and expecting complete Rust output. However, our experiments revealed that SWE-agent exhibited brittle and unreliable behavior in this setting. Furthermore, a developer of OpenHands confirmed via personal communication that such usage falls outside the intended design and is unlikely to yield satisfactory results. Consequently, we focus our evaluation on a pipelined workflow, which demonstrated significantly greater robustness and effectiveness in our experiments.

**Excluded Baselines.** We do not include syntax-directed C-to-Rust transpilers (Immutant Inc.; Zhang et al., 2023; Emre et al., 2021) in our evaluation, as these systems are incompatible with our framework. In particular, they frequently generate *unsafe* Rust code that relies on foreign function interfaces such as libc, thereby violating the safe and idiomatic interface constraints enforced in CRUST-Bench. Similarly, C2SAFERRUST (Nitin et al., 2025), which augments C2RUST with LLM-based post-processing, defaults to generating unsafe code when the target interface cannot be satisfied. Adapting these systems to support our interface-first, safety-preserving evaluation would require significant reengineering and is beyond the scope of this work.

## 5 Results and Analysis

**Single-Shot Transpilation Results.** The first column of Table 4 reports pass@1 test success rates, capturing model performance in a single-shot setting where each model is prompted once per task, without access to compiler or test feedback. Thus, these results evaluate a model's ability to generate correct, safe Rust code on its first attempt. Overall, performance is low across the board: the best-performing model, Claude Opus 4 from Anthropic, passes test cases for only 22% of CRUST-Bench tasks. We omit results for LLaMA-3-8B, which fails to produce any correct solutions.

**Self-Repair Improves Success Rate.** As shown in Table 4, applying iterative self-repair leads to substantial gains in both build and test success rates across all models. With three rounds of Compiler repair, we observe various improvements in build success (as high as 37% gain for o1), and in test pass rates (as high as 13%) compared to the single-shot pass@1 baseline. Appendix D qualitatively discusses the effects of repair. Extending this approach with Test repair, which incorporates failing Rust test cases into the repair loop, yields additional improvements in test success—between 0% and 17% over the corresponding Compiler repair setting.

However, this gain in test correctness comes at the cost of build stability: we observe a 5% to 20% drop in build success when moving from Compiler to Test repair. This degradation can be attributed to the fact that incorporating test case information encourages the model to make more aggressive semantic changes, which may inadvertently introduce new compilation errors—particularly borrowing violations or type mismatches that are difficult to resolve without precise static analysis.

| Model | Config | % projects with error | | | | | | |
|---|---|---|---|---|---|---|---|---|
| | | Mismatch | Borrow | Missing | Unimpl | Trait | Args | Unsafe |
| OpenAI o3 | base | 13 | 21 | 8 | 34 | 4 | 0 | 1 |
| | repair | 9 | 2 | 2 | 27 | 4 | 2 | 0 |
| Claude Opus 4 | base | 28 | 29 | 7 | 13 | 14 | 1 | 6 |
| | repair | 11 | 3 | 2 | 5 | 1 | 3 | 0 |
| OpenAI o1 | base | 30 | 42 | 13 | 14 | 17 | 3 | 0 |
| | repair | 8 | 9 | 2 | 11 | 0 | 1 | 0 |
| Claude 3.7 | base | 15 | 18 | 4 | 44 | 10 | 0 | 0 |
| | repair | 4 | 4 | 4 | 46 | 1 | 1 | 0 |
| Claude 3.5 | base | 24 | 27 | 23 | 37 | 14 | 4 | 0 |
| | repair | 22 | 6 | 16 | 55 | 5 | 5 | 0 |
| o1-mini | base | 46 | 34 | 40 | 28 | 34 | 5 | 0 |
| | repair | 21 | 13 | 10 | 29 | 1 | 6 | 0 |
| GPT-4o | base | 48 | 35 | 25 | 20 | 22 | 4 | 1 |
| | repair | 12 | 20 | 10 | 26 | 6 | 2 | 0 |
| Gemini 1.5 Pro | base | 40 | 24 | 23 | 33 | 17 | 2 | 1 |
| | repair | 18 | 13 | 7 | 35 | 8 | 3 | 1 |
| Virtuoso (Distilled Deepseek V3) | base | 60 | 26 | 19 | 45 | 33 | 12 | 0 |
| | repair | 38 | 23 | 17 | 50 | 15 | 9 | 0 |
| Deepseek-Coder-32B | base | 36 | 13 | 56 | 49 | 22 | 3 | 4 |
| | repair | 28 | 8 | 59 | 36 | 10 | 5 | 2 |
| QwQ-32B-Preview | base | 9 | 2 | 9 | 94 | 5 | 1 | 0 |
| | repair | 13 | 2 | 9 | 92 | 3 | 2 | 0 |
| Qwen-2.5-Coder | base | 7 | 0 | 32 | 76 | 1 | 0 | 1 |
| | repair | 5 | 0 | 9 | 96 | 0 | 0 | 0 |
| Adapted SWE-Agent | | 15 | 17 | 4 | 44 | 8 | 1 | 0 |

Table 5: Error breakdown for different models and configurations

**Agent-Based Debugging.** The last row of Table 4 also presents the results of the pipelined SWE-agent described in the previous section. Compared to the pass@1 baseline (13% test pass rate for Claude 3.7 Sonnet), the pipelined SWE-agent workflow improves performance to 32%, demonstrating a substantial benefit compared the single-shot performance. However, this performance matches, rather than exceeds, that of the Claude 3.7 Sonnet + Test repair strategy, which also reaches 32% after Test repair.

We analyze the behavior of pipelined SWE-agent in Appendix E. We see that pipelined SWE-agent is able to successfully navigate files, edit files, invoke `cargo build` and `cargo test` to build and test the project, and more. However, it does not successfully leverage large numbers of steps to fix errors: many of its steps are simply navigating and reading project files, and most fixed errors are fixed in a relatively small number of steps. This result suggests that, while SWE-agent has broader capabilities — such as reasoning across files, dynamically choosing when to build and test, and autonomously deciding when to submit a solution— it does not leverage these components successfully in our current configuration to outperform the simpler Test repair approach.

**Error Analysis.** To better understand the failure modes, we analyze the distribution of compiler errors in Table 5. These errors were clustered together based on a manual inspection of their description using the `rustc --explain` command. We note that a given project typically exhibits many of these error types at the same time. Based on our manual categorization, we determined that common errors include the following:

- **Mismatch:** These errors arise when the generated Rust code has a type mismatch when calling a function from the tests, another file in the project, or incompatible return types.

- **Borrowing:** These errors arise due to violations of Rust's ownership, borrowing, mutability, and lifetime rules. As an example, such an error might arise if a value is borrowed mutably while it is already borrowed immutably, or if a lifetime annotation is incorrect or unspecified.

- **Missing:** These errors arise due to access to non-existent (or out of scope) variables.

- **Unimpl:** These errors occur due to unimplemented functions or incomplete code segments in the generated output. They often arise when the model exceeds its token budget, leading to truncation before all necessary logic is emitted. In some cases, the model may also omit implementations by stating that certain functions can be implemented "similarly", leaving placeholders or comments instead of actual code.

- **Args:** These errors are observed when a function invocation does not correctly supply the expected number of arguments. (e.g., passes a single argument instead of the required three).

- **Trait-related errors:** These occur due to a failure to implement the necessary traits for custom data types. In Rust, user-defined types often need to implement certain traits to function correctly in the language's type system, and these errors arise if the expected trait is not implemented.

- **Unsafe:** This type of problem occurs when the generated code uses the "unsafe" keyword or a (so-called) unstable Rust feature. We note that these types of problems (e.g., use of "unsafe" keyword) do not necessarily correspond to build failures.

As shown in Table 5, the occurrence of unsafe code is rare. This can be attributed to our prompt design, which explicitly instructs models to avoid using unsafe Rust features.

In contrast, type-related errors (i.e., those in the **mismatch** and **borrow** categories) are quite common. This is expected, given the Rust compiler's strong static guarantees and strict enforcement of type and ownership rules. These errors suggest that models often struggle to reason precisely about lifetimes, mutability, and type compatibility. The prevalence of such errors highlights the potential of incorporating static analysis signals into the fine-tuning process, which could substantially improve LLMs' ability to generate compilable, type-correct Rust code.

Finally, **unimplemented** errors are also widespread. These typically stem from models exceeding their output token limits, leading to truncated or incomplete code. This issue is especially pronounced for models with shorter maximum output token limits. In some cases, the model may omit implementations entirely, inserting placeholder comments or referring to similar functions without emitting code, or violate the interface definitions further contributing to these errors.

## 6 Related Work

**Code Generation Benchmarks.** Prior work has introduced a range of benchmarks for evaluating code generation. Many focus on generating Python code from natural language prompts (Chen et al., 2021; Austin et al., 2021; Jain et al., 2025; Hendrycks et al., 2021), typically framed as short, competition-style problems with test cases. HumanEval (Chen et al., 2021) has been extended to support additional programming languages (Cassano et al., 2023; Athiwaratkun et al., 2023; Orlanski et al., 2023). Other efforts have explored more realistic settings: incorporating external APIs (Yu et al., 2024), class-level code generation (Du et al., 2023), repository-level tasks with inter-file dependencies (Li et al., 2024b), and rigorous testing of HumanEval (Liu et al., 2023a). In contrast, CRUST-Bench evaluates code generation over multi-file C projects with complex dependencies, using code as input rather than natural language, and targets the generation of safe, idiomatic Rust.

**Repository-Level Benchmarks.** Several benchmarks have explored code generation at the repository level. SWE-bench (Jimenez et al., 2024) evaluates a model's ability to resolve GitHub issues using real-world repositories. RepoBench (Liu et al., 2023b) focuses on code completion across entire codebases, while Li et al. (2024a) examine generation across different stages of the software development lifecycle. In contrast, CRUST-Bench shifts the focus from editing or completing code to performing cross-language translation.

**C-to-Rust Transpilation.** Recent work on C-to-Rust transpilation spans both syntactic and learning-based approaches. Tools like C2RUST (Immutant Inc.) focus on syntactic translation, often relying on external libraries like libc and layout-preserving annotations such as #[repr(C)] to maintain compatibility. However, these approaches do not guarantee

memory safety and frequently produce Rust code with unsafe blocks. Other approaches incorporate Rust's ownership model to improve safety (Zhang et al., 2023), though they often still rely on unsafe code. More recent techniques combine syntactic translation with LLM-based post-processing to improve code quality (Nitin et al., 2025; Yang et al., 2024a). While C2SAFERRUST leverages LLMs to clean up C2RUST output, it falls back to unsafe code when constraints cannot be satisfied. In contrast, VERT (Yang et al., 2024a) uses both LLMs and property-based testing to verify semantic equivalence between the source and generated code. Eniser et al. (2024) also adopt LLMs for direct C-to-Rust generation, validating correctness through equivalence checking.

## 7  Conclusion

In this work, we presented CRUST-Bench, a benchmark of 100 C projects with target Rust interfaces and Rust test cases. This benchmark allows us to verify LLM-powered transpilation systems according to three criteria: (1) Do they successfully follow the given interface during transpilation? (2) Does the Rust code compile? (3) Does the code pass the provided Rust test cases? Our results show that even the best approach with state-of-the-art LLMs, OpenAI o3 with iterative repair from both compiler errors and test failures, only succeeds on 48% tasks in the benchmark, leaving significant room for future systems to improve.

## Acknowledgments

This research was conducted within a group supported by the National Science Foundation under awards CCF-1762299, CCF-1918889, CNS-1908304, CCF-1901376, CNS-2120696, CCF-2210831, CCF-2319471, and and by the Defense Advanced Research Projects Agency (DARPA) under Agreement No. HR00112590133. We also thank the All Hands AI team for a discussion on the OpenHands CodeAct agentic framework applied to the C-to-Rust transpilation task.

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

## Appendix

## A  Code Coverage Analysis

We provide additional details on the code coverage analysis conducted over the C repositories in CRUST-Bench. Coverage was measured using gcov (GNU Project) on a subset of 25 benchmarks, selected from projects that successfully built using either make or cmake. For cmake-based builds, some projects linked external files that caused gcov to fail; we did not attempt to resolve these issues across the full benchmark.

Our manual inspection of benchmarks with low coverage revealed common patterns: (1) driver files that invoked core functionality but were themselves untested, and (2) files included for illustrative purposes—such as usage examples or demonstration scripts—that contain calls to library functions but are not executed as part of the test suite, and therefore appear as uncovered in coverage analysis.

In the context of transpilation, we found that models rarely introduced localized errors. For example, when a benchmark included several related functions relying on the same abstract data type, the model typically succeeded or failed on all of them collectively. This suggests that the available test suites, while not exhaustive, are sufficiently representative for evaluating the correctness of transpiled Rust code.

Achieving significantly higher coverage would require writing targeted tests for each function in every benchmark—an effort that entails deep understanding of program intent and edge cases, and is therefore outside the scope of this work.

## B  Greedy self-repair vs sample-and-repair

We describe in more detail our comparison between greedy self-repair vs. sampling multiple outputs from a model and repairing each (balancing the total number of repairs). These are also compared in Olausson et al. (2024), but in a different setting. Both techniques receive feedback in terms of compiler messages.

We conducted a pilot study with 10 tasks in CRUST-Bench, consisting of both single and multiple file projects with a *model calling* budget of 6 per task. The budget was selected to provide a fair comparison between the two techniques. For the sample-and-repair technique, we sample 3 candidates at a temperature of 0.8, used in Olausson et al. (2024), and repair each once. For greedy self-repair, we greedily sampled a single candidate and repaired it up to 5 times with a temperature of 0. We tested this with Claude-3.5 and GPT-4o. For both LLMs, we found that greedy self-repair outperformed sample-and-repair. We attribute this to two factors: (1) The compiler may not surface all errors at once and multiple iterations of self-repair are required to address all errors, and (2) addressing errors might introduce new errors, which may require fixing in subsequent iterations.

For our full experiments, we found that 3 rounds of greedy self-repair were sufficient. Beyond three rounds we did not see substantial improvements.

## C   Prompts

**Prompt for Transpilation**

```
You are an expert at converting C To Rust.
You will be provided with C source files in the format:

  {{filename.c / filename.h}}
  ```c
  // Input C code
  ```

  I need you to transpile the provided code files from C to Rust, with the
  following instructions that you MUST follow:
  - Each C file I provide MUST have the corresponding transpiled Rust file.
  - You will also be given the Interface files with function signatures
  and return types. You MUST conform to the specification given in the
  interface definitions.
  - Each transpiled Rust file MUST compile.
 - The transpiled Rust code MUST be observationally equivalent to the C code.
  - The transpiled Rust code MUST compile successfully.
 - The transpiled Rust code MUST NOT contain Foreign Function Interface calls,
  such as the libc library.
  - The transpiled Rust code MUST NOT contain unsafe blocks.
  - All imports in the rust project (except main.rs) MUST be in the
  following format -
    ```rust
    use crate::file_name::module;
    ```

  - Imports in main.rs should be done in the following format:
    ```rust
    use project_name::file_name::module;
    ```

 - You MUST ensure that you include the required files that are referenced in
  each rust file.
  Please think step-by-step and return your final solution for each
  transpiled file in the following format:

  {{filename.rs}}
  ```rust
  // Generated Rust Code
  ```
```

**Prompt for Compiler Repair**

```
You are an expert at repairing Rust Code.
You will be provided with Rust code in the format:
  {{filename.rs}}
  ```rust
  // Input Rust code
  ```

After the Rust code, you will be given the errors obtained from the compiler
corresponding to the Rust code.

I need you to repair the provided Rust files,
with the following instructions that you MUST follow:
  - You MUST produce the entire file when repairing it and
  not just the intended change.
  - You MUST NOT change the function signatures.
  - You MUST address each error by reasoning about it.
  - Each error MUST be solved using safe Rust code.
  - The transpiled Rust code MUST compile successfully.
 - The transpiled Rust code MUST NOT contain Foreign Function Interface calls,
  such as the libc library.
  - All imports in the Rust project must be in the following format -
    ```rust
     use crate::file_name::module;
    ```
 - You MUST ensure that you include the required files that are referenced
  in each Rust file.
 - You MUST ensure not to change the function signatures and return types of
  the functions when you are performing repairs.

 Please think step-by-step and return your final solution for each
 transpiled file in the following format:

 {{filename.rs}}
 ```rust
 // Generated Rust Code
 ```
```

**Prompt for Test Repair**

```
You are an expert at Rust.
You will be provided with Rust code in the format:
  {{filename.rs}}
  ```rust
  // Input Rust code
  ```
After the Rust code, you will be given the test files that were executed on
the Rust source code in the format
  {{filename.rs}}
  ```rust
  // Input Rust code
  ```
After that, you will be provided with the test failures obtained from the
compiler corresponding to the Rust code.
I need you to repair the provided Rust files, with the following instructions
that you MUST follow:
  - You MUST produce the entire file when repairing it and not just
  the intended change.
  - You MUST not change the test code at all. You must only make fixes to the
  Rust source files.
  - You MUST NOT change the function signatures.
  - You MUST address each failure by reasoning about it.
  - Each test failure MUST be addressed using safe Rust code.
  - The corrected Rust code MUST compile successfully.
 - The corrected Rust code MUST NOT contain Foreign Function Interface calls,
  such as the libc library.
  - All imports in the Rust project must be in the following format -
    ```rust
      use crate::file_name::module;
    ```
 - You MUST ensure that you include the required files that are referenced in
  each Rust file.
  - You MUST ensure not to change the
 function signatures and return types of the functions when you are performing
  repairs.
 Please think step-by-step and return your final solution for each transpiled
  file in the following format:

  {{filename.rs}}
  ```rust
  // Generated Rust Code
  ```
```

## System instruction for SWE Agent

```
SETTING: You are an autonomous programmer, and you're working directly in
the command line with a special interface. Your task pertains to solving
errors in a repository with Rust code.
You must solve the problem by adding Rust code to the repository. You can
use the special interface to navigate and edit files. You can also use
any bash commands to help you solve the problem.
You are provided with the Rust files and the associated Rust test files.
Your implementations should go in the src directory. You can use the
`cargo build` command for building the project and `cargo test`
for running the tests.

  The special interface consists of a file editor that shows you {{WINDOW}}
lines of a file at a time.
In addition to typical bash commands, you can also
Use the following commands to help you navigate and edit files.

  COMMANDS:
  {{command_docs}}

  RESPONSE FORMAT:
  Your shell prompt is formatted as follows:
  (Open file: <path>) <cwd> $

  You need to format your output using two fields: discussion and command.
  Your output should always include _one_ discussion and _one_ command field
EXACTLY as in the following example:
DISCUSSION
  First, I'll start by using ls to see what files are in the current directory.
  Then maybe we can look at some relevant files to see what they look like.
  ```
  ls -a
  ```

You should only include a *SINGLE* command in the command section and
then wait for a response from the shell before continuing with more
discussion and commands. Everything you include in the DISCUSSION section
will be saved for future reference.
If you'd like to issue two commands at once, PLEASE DO NOT DO THAT!
Please, instead first submit just the first command, and then after receiving
a response, you'll be able to issue the second command.
You're free to use any other bash commands you want (e.g., find, grep, cat,
ls, cd) in addition to the special commands listed above.
  However, the environment does NOT support interactive session commands
(e.g., python, vim), so, please do not invoke them.
```

---

**Problem statement provided to SWE Agent**

```
You are provided with a Rust project that must pass all the tests. In case the
code contains errors, you must repair the provided Rust files such that all
test cases pass.
You first run `cargo build` to determine what errors exist in the project.
In case you don't find any errors, you try running tests defined in the
project using `cargo test`. In case you find any errors when executing
`cargo build` and `cargo test`, you must address them by changing the Rust code
in the src/ folder of the project. Incase you don't find any errors, you submit
the project.
You must ensure that you follow the following instructions:
  - You must not change the function signatures and return types of the
  functions when you are performing repair on a file.
  - You must address each error by carefully reasoning about it.
  - Each error must be solved using safe Rust code.
 - The transpiled Rust code must not contain Foreign Function Interface calls,
  such as the libc library.
  - All imports in the Rust project must be in the following format -
    ```rust
     use crate::file_name::module;
    ```
 - You must ensure that you include the required files and constants that are
  referenced in each Rust file.
 - You must ensure that you do not change the test code in the src/bin folder.
Please think step-by-step and resolve the issue.
```

---

## D  Analysis of the Self-Repair Pipeline

We observe that self-repair is effective at mitigating a range of errors encountered during the initial transpilation pass. For Claude 3.7 Sonnet in particular, we note the following trends:

- **Unimplemented functions and imports:** The model generally fails to recover from unresolved function and import errors during self-repair. This limitation arises from the fact that the LLM is more adept at editing existing code than synthesizing entirely new logic during repair rounds. In 2% of benchmarks, the self-repair process increased the number of unimplemented functions, further exacerbating these errors.

- **Borrow-checker errors:** Borrowing-related errors are reduced by 75% after applying compiler-guided repair, suggesting that these errors are relatively tractable when explicit compiler feedback is available.

- **Trait-related errors:** Errors related to missing trait implementations are reduced by 90%, indicating that trait errors are particularly amenable to iterative correction.

- **Type mismatches:** Type errors are cut by 50% in the first round of repair but plateau in subsequent rounds, suggesting diminishing returns on further iterations for this category.

- **Mutability issues:** We observe that attempts to fix type and borrow errors sometimes introduce new mutability-related issues, which may or may not be corrected in later repair passes.

While self-repair improves many error types, we also identify cases where it fails to yield performance gains or introduces new issues. In particular, we observe that attempts to address specific errors often cascade into other categories of failure.

For example, in the multi-file project impcheck, the model initially omits the hash.rs file containing the core HashTable implementation. In the first repair round, it adds hash-related functions but omits a required Clone trait implementation. The second round corrects this omission, but introduces a new error: the compiler reports that the capacity method is missing. In the third round, the LLM adds this method, but the implementation mishandles

ownership, leading to borrow-checker violations. This case illustrates how self-repair can make steady progress, yet ultimately fail due to deeper semantic issues related to Rust's ownership model.

A similar pattern appears in the graph-recogniser project. The initial translation includes borrowing errors tied to a custom Graph type defined in the library. Despite multiple repair attempts, the model is unable to satisfy the ownership constraints required for the correct use of this type. Additionally, the model fails to incorporate a custom macro used for hashing graph nodes—highlighting its difficulty in reasoning about domain-specific constructs that are not easily inferred from compiler error messages alone.

## E   Results using SWE-Agent

**Characterization of SWE-Agent Behavior.**   The SWE-Agent framework consists of an orchestrating LLM that executes a sequence of structured actions to iteratively generate patches for a target repository. A key question in our analysis is: what types of actions does SWE-Agent take in attempting to fix transpilation errors?

According to Table 4 in the SWE-Agent paper (Yang et al., 2024c), the system can invoke standard Bash commands in addition to eleven custom actions: seven related to file viewing and search, three dedicated to editing, and one final action to submit the current patch for evaluation.  In our case, evaluation involves compiling the project and running the associated test suite.

From qualitative analysis, we observe a consistent pattern in SWE-Agent's behavior:

- It begins by exploring the repository, viewing both source and test files using a combination of Bash navigation and custom file-view actions.
- Once sufficient context is gathered—typically after observing all relevant files—it issues the first cargo build command. On average, this occurs at the $12^{\text{th}}$ step.
- If compilation errors are encountered, SWE-Agent identifies the problematic files and applies targeted edits.  When changes span multiple files, it navigates across them accordingly.
- After each set of edits, it re-runs cargo build, repeating this process until the project builds or the execution budget is exhausted.
- Upon a successful build, SWE-Agent executes cargo test. If tests fail, it inspects the output and attempts further edits to address the test failures.
- Once all test cases pass, the agent submits the final patch for the repository.

Figure 6 presents a quantitative breakdown of SWE-Agent's behavior under a $2 execution budget.  The agent performs a nontrivial number of navigation actions, as indicated by frequent directory listings. However, it rarely uses the scroll operation, suggesting that it often avoids reading beyond the first 100 lines of any file.

We also observe that SWE-Agent typically edits a moderate number of lines and invokes cargo build and cargo test multiple times per project—regardless of success—highlighting its reliance on interactive feedback from the build and test system.

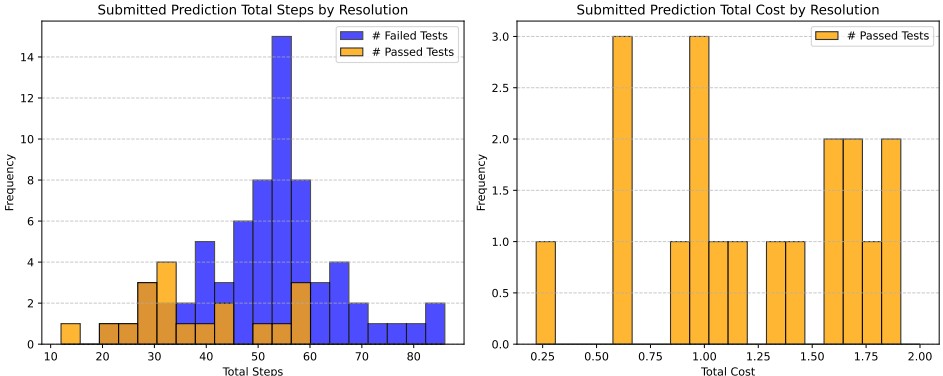

Figure 3: Statistics of pipelined SWE-agent with a cost budget of $2. Left: Distribution of steps taken until submission/exit. We see that a majority of resolved test failures (∼80%) are addressed within the first 50 steps of SWE-agent, showcasing early converge when failures are recoverable. Right: Distribution of cost required to fix test failures successfully.

**Comparison Across Cost Budgets.**
Table 6 compares SWE-Agent performance under varying cost budgets. Even with a $1 budget, SWE-Agent outperforms the baseline pass@1 setting, demonstrating that modest interaction can yield meaningful improvements.

| Cost Budget | Test(Pass@1) |
|---|---|
| $1.00 | 14 |
| $2.00 | 32 |
| $4.00 | 21 |

Table 6: Test pass rates of SWE-agent at different cost budgets.

However, performance plateaus at the $2 mark, with no observable gains between $2 and $4. This suggests diminishing returns with additional budget, likely due to the agent exhausting meaningful actions within the first few iterations.

Figures 3 and 4 help explain this plateau in performance. When SWE-Agent is successful, it typically requires only a small number of steps, often completing the task early in the interaction sequence. This observation is consistent with the findings reported in Section 5.2 of the original SWE-Agent paper (Yang et al., 2024c).

Figure 5 further illustrates this behavior by analyzing the frequency of cargo build and cargo test commands at different budget levels. Between the $1 and $2 budgets, we observe a substantial increase in build and test invocations. However, increasing the budget from $2 to $4 does not lead to additional successful builds, indicating that the agent often exhausts its effective repair strategies within the lower budget range.

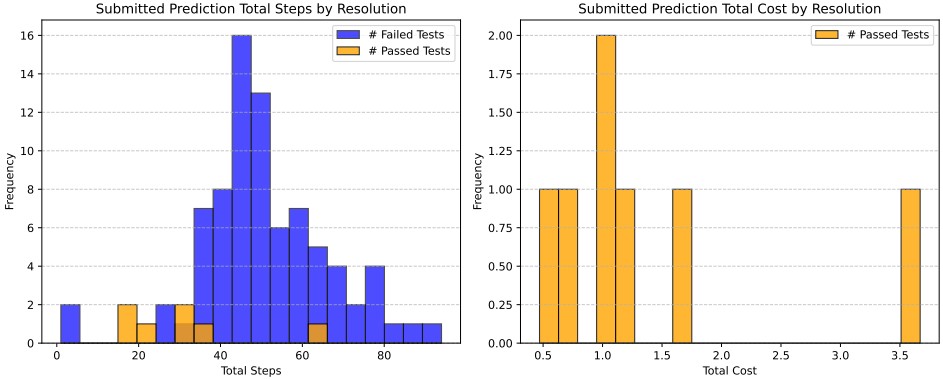

Figure 4: Statistics of pipelined SWE-agent with a cost budget of $4. Left: Distribution of steps taken until submission/exit. We see that a majority of tasks are addressed within the first 40 steps of SWE-agent, again showcasing early convergence, even with a higher cost, in cases where errors are recoverable. Right: Distribution of cost required to fix tests successfully. Only 1 task takes over $3.5 to be addressed.

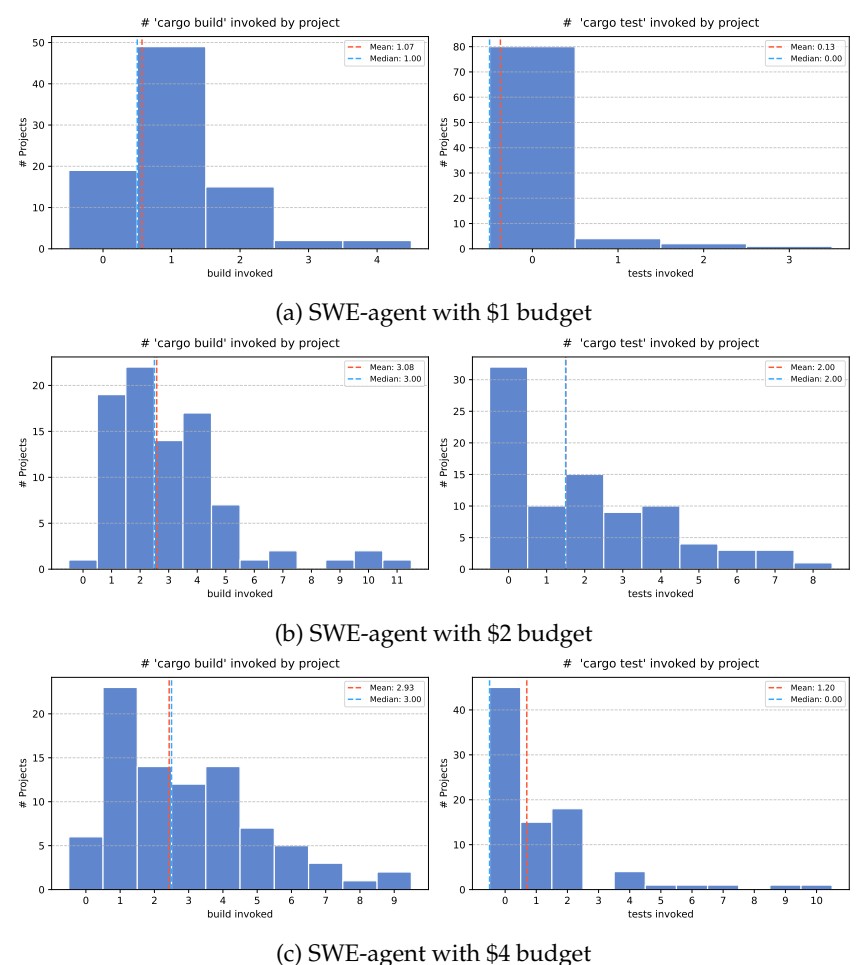

Figure 5: Analysis of SWE-agent build and test command invocations across different budget levels. We note that the model invokes the build command more as the cost budget is increased. The average number of test invocations remains the same.

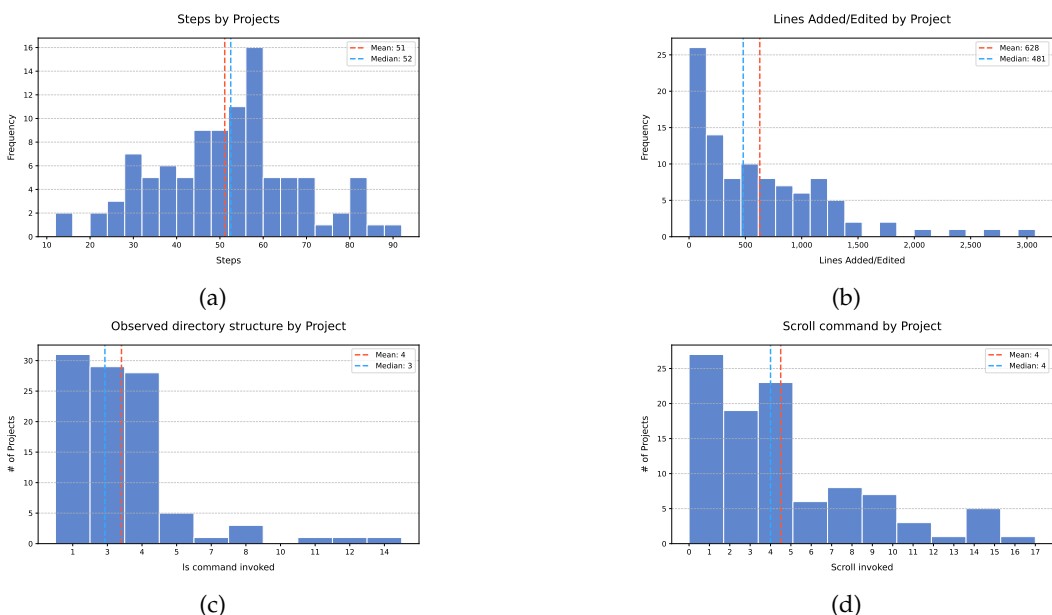

Figure 6: Analysis of commands and their frequency across projects.

