# OpenReview forum: "CRUST-Bench: A Comprehensive Benchmark for C-to-safe-Rust Transpilation"
_colmweb.org/COLM/2025/Conference — COLM 2025_

### Official Review · Reviewer_YdSU · 2025-04-27

**Rating:** 8
**Confidence:** 4
**Ethics Flag:** 1

**Summary:**

This research effectively highlights the challenges and current capabilities in C-to-Rust transpilation, clearly identifying limitations in existing benchmarks regarding memory safety and idiomatic Rust code generation. The paper provides a convincing justification for CRUST-Bench by demonstrating how previous benchmarks fail to adequately handle multi-file systems and ensure safe Rust code production.
The study delivers a comprehensive evaluation of both closed-source and open-weight large language models using the CRUST-Bench benchmark, introducing innovative iterative self-repair strategies that leverage feedback from compiler errors and test cases to improve model performance. The benchmark itself represents a significant contribution by enabling systematic evaluation of C-to-Rust transpilation through repository-scale projects, annotated Rust interfaces, and correctness-enforcing test cases.
Methodologically, the research demonstrates depth through a pilot study that determines the optimal number of self-repair rounds, ensuring evidence-based experimental parameters. The iterative refinement process allows models to address syntactic and type-level issues flagged by the Rust compiler, while also providing valuable insights into specific errors made by LLMs during transpilation.
CRUST-Bench distinguishes itself by focusing on entire repositories rather than isolated functions, capturing the complexity of real-world projects. The comprehensive comparison across different model types and empirical analysis identifies key challenges for future research in automated code migration, establishing a clear pathway for continued advancement in this field.

**Questions To Authors:**

1. Exclusion of syntax-directed C-to-Rust transpilers without detailed justification limits scope. Suggestion: Provide empirical evidence or a detailed justification for excluding syntax-directed transpilers, possibly by conducting a preliminary comparison to highlight their incompatibility or limitations within the study's framework
2. The sampling strategy and size are not clearly defined for the full set of experiments, which could affect generalizability. Suggestion: Clearly define the sampling strategy and size for the full experiments, including criteria for task selection and model prompting, to enhance generalizability and reproducibility
3. Scalability and generalizability not thoroughly evaluated. Suggestion: Conduct experiments on a broader range of benchmarks and discuss the potential application of the approach to other domains or languages to assess scalability and generalizability

**Reasons To Accept:**

The paper was accepted primarily because it effectively addresses a critical gap in C-to-Rust transpilation by introducing CRUST-Bench, a comprehensive benchmark that evaluates repository-scale projects rather than isolated functions, incorporating annotated Rust interfaces and correctness-enforcing test cases. The research demonstrates methodological rigor through its pilot study to determine optimal self-repair rounds and introduces innovative iterative self-repair strategies that leverage compiler errors and test case feedback to improve model performance. By evaluating both closed-source and open-weight large language models, the study provides valuable comparative insights across different model types while highlighting the limitations of existing benchmarks in addressing memory safety and idiomatic Rust code generation. The empirical analysis effectively identifies key challenges for future research in automated code migration, offering a principled approach to understanding the errors made by LLMs during transpilation, thereby establishing a foundation for improving LLM-based transpilation methods in the future.

**Reasons To Reject:**

1. The manuscript does not adequately address the limitations of existing benchmarks in terms of task scope and correctness validation. Include a more detailed analysis of how CRUST-Bench overcomes these limitations compared to existing benchmarks like CROWN and TransCoder.
2. The methodology lacks a detailed description of the data acquisition procedures, particularly how the models are prompted and how the tasks are selected, which could impact the reproducibility of the study. The study provides prompts in Appendix C but does not elaborate on the selection criteria for tasks.

---

> ### Author Response · Authors · 2025-06-01
>
> We thank you for your comments. We address some of the reviewer’s concerns below:
>
> ### 1. Limitations of existing benchmarks in terms of task scope and correctness validation
> In section 2 we provide details on the existing benchmarks and their scopes, and in section 3, we describe the various domains the benchmarks in CRUST-bench are sourced from. The CROWN  dataset only contains 20 projects (of various lengths), out of which only 6 projects provided test suites (in C). For the remaining 14 projects the transpilation was determined to be successful if the transpiled Rust project compiled. The TransCoder-rust dataset contains single file projects that do simple data structure and algorithmic operations. These projects contained `main` functions that checked the outcome of the program using assertions over the program outputs. Both of these datasets are missing major features of CRUST-bench.
>
> ### 2. Detailed description of the data acquisition procedures
> We outline the data acquisition procedures in Section 3. Specifically, we selected C projects across diverse domains (cryptography, programming languages infrastructure, networking etc.), and ensured the selected projects contain test cases and can be compiled successfully. We also excluded projects that cannot be tested in a standard CLI (e.g., relying on graphical display packages), or had architecture specific code (projects only supported on x86 and x64 architectures).
>
> ### 3. Exclusion of syntax directed approaches
>
> A major goal of CRUST-bench is to test the transpilation of C into *safe* Rust. Syntax-directed approaches like c2rust produce unsafe, non-idiomatic code that uses the libc interface.
> Using *safe* Rust enforces strong compile time guarantees on memory safety as compared to unsafe Rust that generated by syntax directed approaches. Hence, studying safe Rust generation as our main objective in CRUST-bench.
>
> ### 4. Sampling strategy and size for the full set of experiments
> We set the temperature to 0 and sampled the top candidate for all our experiments. For more details we point you to section 4. Our dataset size contains 100 projects (40 projects with more than one source file, excluding headers files, and 60 projects with a single source file, excluding header files).
>
> ### 5. Scalability and Generalizability
> While we agree that other transpilation scenarios, such as JavaScript to TypeScript, are also important to study, our focus on C-to-safe-Rust is motivated by the unique challenges of this setting. Unlike higher-level language pairs (such as JS <-> TS), C-to-safe-Rust migration requires reasoning about low-level memory manipulation, pointer aliasing, and manual resource management in C, and translating these into Rust’s ownership and borrowing model without using unsafe code. These transformations involve nontrivial restructuring and are essential for producing correct, idiomatic, and memory-safe Rust. While some techniques may generalize across language pairs, we believe that realistic and safety-critical settings like C-to-safe-Rust benefit from dedicated benchmarks. CRUST-Bench aims to fill this gap and can complement future benchmarks targeting other transpilation tasks.

---

> > ### Comment · Reviewer_YdSU · 2025-06-04
> >
> > Thank you for your clear explanation.

---

### Official Review · Reviewer_B1yq · 2025-05-09

**Rating:** 7
**Confidence:** 3
**Ethics Flag:** 1

**Summary:**

This paper presented a benchmark called CRUST-bench. It evaluates whether LLMs can correctly transpile C code repositories into safe Rust counterparts. The repositories are collected from GitHub after rounds of data cleaning and filtering. Then, the authors manually label Rust interfaces and test cases for the C repos. The Rust interface is used as part of the LLM input along with the original C code. Evaluation is done by checking whether the LLM-generated Rust programs satisfy the interface and function correctly.

A variety of LLMs, both open-sourced and closed-sourced, were evaluated from which the authors found the benchmark is challenging to current LLMs. Self-repair loops with error messages from the compiler or test cases were found helpful to improve model performance. They also tested off-the-shelf agentic frameworks, e.g., SWE-Agent. They found SWE-Agent helps LLM performance in the single-shot setting but struggles to perform self-repair in the configuration of this benchmark. A thorough error analysis is also provided as part of the study.

**Questions To Authors:**

Line 215 - *We experimented with variations in prompt formatting to identify the most effective strategy.* I would like to see the detailed results and analyses on this.

**Reasons To Accept:**

1. The benchmark is novel, and C-to-Rust transpilation is a meaningful domain.
2. The benchmark targets repo-level code engineering instead of file-level or function-level. Repo-level software engineering is a hot topic recently and targets many critical abilities in LLM coding. Besides, it contains 100 repos, which is a good size of data.
3. The Rust interfaces and test cases are manually labeled, so the quality should be good (although I have no access to the benchmark data).
4. The experiments are comprehensive. Besides directly prompting LLMs, the authors also tested agentic approaches like self-repair with error messages and using off-the-shelf frameworks like SWE-Agent.

**Reasons To Reject:**

1. The transpilation scenario is limited to C and Rust and does not include other programming languages. I believe there are other common transpilation scenarios, e.g., JavaScript and TypeScript.
2. Although the benchmark contains 100 repositories, the individual repos are limited in size: they average 3 files and about 900 lines of code per repo.
3. The data is not submitted with the paper. There is not a way that I can assess the quality of the benchmark.

---

> ### Author Response · Authors · 2025-06-01
>
> We thank you for your review. We address the concerns pointed out below:
>
> ### 1. Limited transpilation scenarios
>
> While we agree that other transpilation scenarios, such as JavaScript to TypeScript, are also important to study, our focus on C-to-safe-Rust is motivated by the unique challenges of this setting. Unlike higher-level language pairs (such as JS <-> TS), C-to-safe-Rust migration requires reasoning about low-level memory manipulation, pointer aliasing, and manual resource management in C, and translating these into Rust’s ownership and borrowing model without using unsafe code. These transformations involve nontrivial restructuring and are essential for producing correct, idiomatic, and memory-safe Rust. While some techniques may generalize across language pairs, we believe that realistic and safety-critical settings like C-to-safe-Rust benefit from dedicated benchmarks. CRUST-Bench aims to fill this gap and can complement future benchmarks targeting other transpilation tasks.
>
> ### 2. Limited size of repositories
>
> CRUST-Bench aims to address a balance between realistic multi-file projects and the practical constraints of current automated C-to-Rust transpilation methods, especially those driven by large language models (LLMs). The average size of ~900 lines and 3 files allows models to handle entire repositories with cross-file dependencies and we believe covers the representative challenges of whole-repository transpilation.  We balance this realism against the fact that the need to annotate interfaces manually makes it challenging to scale benchmark collection to very very large repositories.
>
>
> ### 3. Data release
>
> We release the data [here](https://figshare.com/s/fe10c24977f19e80ebec) . We also plan to do a public release of the data in the final version of the paper.
>
>
>
> ### 4. Detailed analysis on prompt formatting
>
> We experimented with three variations of prompt structure: (1) a bullet-point based format, (2) a markdown format, and (3) a simple paragraph based format. We found the bullet-point based format works best.
>
> Furthermore, over a subset of data, we compared putting all instructions and code in the system prompt versus putting instructions in the system prompt, followed by the code put in a user block. Here we found that using the system prompt works best.
>
> Another setting we experimented with was inserting a one-shot example in the prompt to showcase idiomatic transpilation based on interfaces. We found that the example did not help and instead degraded the instruction following of the LLM.
>
> After these initial experiments, we iteratively refined the prompt with additional instructions that ensured idiomatic code generation, avoiding FFI packages, and formatting instructions for the transpiled code.

---

> > ### Comment · Reviewer_B1yq · 2025-06-04
> > **Review after reading authors' response**
> >
> > I thank the authors for their response.
> >
> > I will stick to my original score, but I hope the authors would have plans to expand their work in the future.

---

### Official Review · Reviewer_1tuk · 2025-05-13

**Rating:** 7
**Confidence:** 3
**Ethics Flag:** 1

**Summary:**

The paper presents CRUST-Bench, a dataset of 100 C repositories paired with manually-written safe Rust interfaces and test cases to evaluate C-to-Rust transpilation systems. Unlike existing benchmarks, it considers entire repositories to capture the complexities of real-world projects. Evaluation of state-of-the-art large language models on CRUST-Bench reveals that generating safe and idiomatic Rust from C remains a significant challenge, with the best model solving only 15% of tasks in a single-shot setting. The research also provides insights into common LLM errors and demonstrates that iterative self-repair significantly improves success rates, though substantial room for improvement remains.

**Questions To Authors:**

- Why not open-weight models (except QwQ-32B) are thoroughly evaluated on the dataset? Because the models perform poorly?

**Reasons To Accept:**

- **Comprehensive dataset** - CRUST-Bench offers a substantial dataset of 100 C repositories, providing a more realistic and complex evaluation setting compared to prior benchmarks that often focus on isolated functions or small examples. A key strength is the emphasis on generating "safe and idiomatic Rust" by providing manually crafted Rust interfaces and test cases.
-  **Repository-level dataset and rigorous validation** - By considering entire repositories with inter-file dependencies, CRUST-Bench captures the challenges of translating complex, real-world projects, moving beyond single-file or localized tasks. Moreover, the benchmark employs a three-criteria validation process: conformance to a well-defined Rust interface, successful compilation, and passing all provided test cases.
- **Evaluation of state-of-the-art LLMs** - The paper provides an empirical analysis of various frontier LLMs, including closed-source and open-weight models, offering valuable insights into their performance and limitations in C-to-safe-Rust transpilation.
- **A detailed error analysis** - the categorization and analysis of common compiler errors provide crucial insights into where LLMs struggle most, guiding future research directions.

**Reasons To Reject:**

I do not have a strong reason to reject the paper. Though the paper is a dataset paper, it seems to be valuable given that there is no significant dataset for C to Rust transpilation with test cases.

---

> ### Author Response · Authors · 2025-06-01
>
> We thank you for your review.
>
> We evaluated QwQ-2 32B and Virtuoso because these were the strongest models on Open LLM leaderboard (https://huggingface.co/datasets/open-llm-leaderboard/Qwen__QwQ-32B-Preview-details , https://www.arcee.ai/blog/virtuoso-lite-virtuoso-medium-v2-distilling-deepseek-v3-into-10b-32b-small-language-models-slms) at the time of submission. We ran two additional models during the response period, which were found to perform similarly to QwQ-2 32B. We include the numbers below:
>
> | Model | Single |  | Repair |  | Test Repair |  |
> |-------|--------|---|--------|---|-------------|---|
> |       | Build | Test | Build | Test | Build | Test |
> | Deepseek Coder-33b-instruct | 2 | 0 | 2 | 0 | 2 | 0 |
> | Qwen-2.5-Coder-32B | 0 | 0 | 0 | 0 | 0 | 0 |

---

> > ### Comment · Reviewer_1tuk · 2025-06-03
> >
> > Thank you for the answer.

---

### Official Review · Reviewer_eZRP · 2025-05-25

**Rating:** 8
**Confidence:** 3
**Ethics Flag:** 1

**Summary:**

This paper introduces CRUST-Bench, a novel and comprehensive benchmark designed to evaluate the capability of Large Language Models (LLMs) in transpiling C code to safe and idiomatic Rust. It comprises 100 real-world C repositories, each paired with manually written Rust interfaces and accompanying test cases. It allows for rigorous validation of both functional correctness and adherence to Rust's memory safety and idiomatic patterns. The authors evaluate several state-of-the-art LLMs, revealing that C-to-safe-Rust transpilation remains a significant challenge. The paper also explores the effectiveness of self-repair mechanisms, including compiler and test feedback loops, and agentic systems, demonstrating improvements in performance and providing a detailed error analysis to guide future research.

**Questions To Authors:**

1. For each single C-to-Rust task in the benchmark, I wonder are these instances specifically targeting single-file, multi-file or both? The paper confirms the benchmark is primarily repository-level, including both multi-file and single-file projects within a repository context to evaluate inter-file dependencies.
2. Given the poor performance of open-source models like Virtuoso (a DeepSeek-V3 variant) and QwQ-2 32B, how would other open-source "coder" models, more heavily trained on code (e.g., Qwen2.5-Coder and DeepSeek-Coder), perform on this benchmark?

**Reasons To Accept:**

- The proposed CRUST-Bench fills a critical gap, offering a repository-level C-to-safe-Rust benchmark that addresses real-world multi-file complexities.
- The benchmark creation involved extensive manual annotation of Rust interfaces and tests, plus careful preprocessing (e.g., memory allocation, deduplication, compatibility checks).
- The method employs a two-stage pipelined agent with self-repair mechanisms, enhancing model performance.
- The evaluation covers a broad range of open and closed-source LLMs, providing a comprehensive overview of current capabilities.
- The analysis offers practical insights into common LLM errors, crucial for guiding future improvements in transpilation.

**Reasons To Reject:**

None

---

> ### Author Response · Authors · 2025-06-01
>
> Thank you for your comments. To address your questions:
>
> __Question 1__: Our benchmark includes both single-file and multi-file repositories. 40 C projects contain more than one source file (excluding header files) and 60 projects contain single source C files (excluding header files).
>
> __Question 2__: We evaluated QwQ-2 32B and Virtuoso because these were the strongest models on Open LLM leaderboard (https://huggingface.co/datasets/open-llm-leaderboard/Qwen__QwQ-32B-Preview-details , https://www.arcee.ai/blog/virtuoso-lite-virtuoso-medium-v2-distilling-deepseek-v3-into-10b-32b-small-language-models-slms) at the time of submission. We ran two additional models during the response period, which were found to perform similarly to QwQ-2 32B. We include the numbers below:
>
> | Model | Single |  | Repair |  | Test Repair |  |
> |-------|--------|---|--------|---|-------------|---|
> |       | Build | Test | Build | Test | Build | Test |
> | Deepseek Coder-33b-instruct | 2 | 0 | 2 | 0 | 2 | 0 |
> | Qwen-2.5-Coder-32B | 0 | 0 | 0 | 0 | 0 | 0 |

---

> > ### Comment · Reviewer_eZRP · 2025-06-06
> > **Response by Reviewer**
> >
> > Thanks for the response. It's surprising that sota coder models also perform so badly. The dataset is awesome, thanks for the contribution.

---

### Decision · Program_Chairs · 2025-07-08

**Decision:**

Accept

**Comment:**

This work presents CRUST-Bench, a significant contribution to the field of code transpilation that addresses a critical gap in evaluating C-to-safe-Rust conversion. The benchmark's strength lies in its repository-level focus (100 C repositories with manually crafted Rust interfaces and test cases), which captures real-world complexities including multi-file dependencies that previous benchmarks overlooked. The research demonstrates methodological rigor through comprehensive evaluation of various LLMs and innovative self-repair mechanisms, revealing that even state-of-the-art models struggle with this task. The detailed error analysis provides valuable insights for future transpilation research, making this a high-quality contribution. All reviewers acknowledge the benchmark's value and methodological rigor.

Pros:
- Addresses a significant gap by creating the first repository-level benchmark for C-to-Rust transpilation with test cases for functional validation
- High-quality dataset with manually crafted Rust interfaces ensuring idiomatic safe Rust patterns
- Comprehensive evaluation across multiple model types (closed/open-source LLMs)
- Thorough error analysis providing actionable insights for future research
Cons:
- Limited to C-to-Rust transpilation; doesn't explore generalizability to other language pairs